

# Soil lacquer peel DIY: simply capturing beauty

Cathelijne R. Stoof[1], Jasper H.J. Candel[1], Laszlo van der Wal[1], Gert Peek[1]

[1] Soil Geography and Landscape Group, Wageningen University, PO Box 47, 6700 AA Wageningen, The Netherlands

*Correspondence to*: Cathelijne R. Stoof (cathelijne.stoof@wur.nl)

**Abstract.** Visualization can greatly benefit understanding of concepts and processes, which in soil science and geology can be done using real life snapshots of soils and sediments in lacquer peels and glue peels. While it may seem complicated, anyone can make such a soil peel for use in classrooms, public places, homes and offices for teaching, outreach, decoration and awareness. Technological development has considerably simplified the making of soil peels, but this methodological innovation has not been described in the literature. Here, we report on a thoroughly tested and simple method for taking peels
of sandy soils using readily available tools and materials. Our method follows the main previously published steps of preparing a soil face, impregnating the soil face with a fixation agent in the field, extracting the resulting peel and mounting it on a wooden panel. Yet instead of using lacquers and thinning agents, we use strong though flexible contact adhesive (glue), which has the major advantage that it no longer requires use and mixing of toxic chemicals in the field or reinforcement of the peel to prevent breaking. Moreover, the preservation potential is much higher than with the old method. This new twist to old
methods makes creating of soil peels more safe, simple and successful, and a thereby true DIY (do it yourself) activity. The resulting increased accessibility of making soil and sediment peels can benefit research, teaching, and science communication and can thereby bring the value and beauty of the ground below our feet to students, schools, policy makers, and the general public.

## 1 Introduction

Attention for soils is increasing around the world, in part due to strong initiatives on soil health (Stott and Moebius-Clune, 2017;Schindelbeck et al., 2008) and soil carbon (4‰, Minasny et al., 2017), and explicit articulation of how soils can help achieve the United Nations Sustainable Development Goals (Keesstra et al., 2016;Bouma and Montanarella, 2016). The relevance of soils lies in the valuable beauty of soils: their multidisciplinary functions and benefits (Brevik et al., 2015;Dominati et al., 2010) and thereby their basis for life, in a world where soils are under threat (Montanarella et al., 2016).
Capturing this beauty in monoliths or soil lacquer peels can bring soils *to* life for education and outreach (Van Baren and Sombroek, 1981;Lawrie and Enman, 2010) or as a form of art (Feller et al., 2015;Breaker, 2013). While it is often thought to be quite challenging to capture soils, a simple twist to an old method now makes the creation of soil peels a surprisingly simple Do-It-Yourself (DIY) activity for scientists, educators and the general public.

Soils and sediments can be fixated in two distinct ways: using peels and monoliths. Both methods rely on impregnation of a
30 soil face with a fixation agent (such as lacquer, resin, or glue), and their final product is typically mounted on a wall for study





of undisturbed soil layers and characteristics, or simply for decoration. Peels and monoliths are used to record and illustrate a range of different features in soils, such as differences between soil types, soil processes (e.g. weathering, gley, eluviation and illuviation of clay, iron, and organic acids (Fig. 1a)), human impacts (Fig. 1b) as well as biological activity such as plant rooting patterns, burrowing of soil fauna, and bioturbation. Sedimentological and geological processes can also be captured,

such as cryoturbation, fluvial and aeolian layering (Fig. 1c), frost wedges (Fig. 1d), and faults (Fig. 1e). And finally, peels can show the splendid colours present in soils and sediments (Fig. 1a-f). These natural snapshots of the subsurface are an effective way to inspire people about soils (Megonigal et al., 2010a) and geology, and are used around the world by museums, universities, schools and institutes (Table 1) for teaching and outreach on the value of soils, the processes occurring in soils, effects of management, and other factors. Interestingly, these soil profiles are also used for testing knowledge of soils in job

interviews (personal communication, Jacqueline Hannam). Peels and monoliths allow comparison of soils inside a classroom or museum environment without the need for students or visitors to travel to see a variety of soils. Consequently, soil science education at Wageningen University, The Netherlands, strongly relies on a collection of ~150 lacquer peels for teaching purposes – despite the fact that this university is intentionally strategically located in an area where soil variability is high (van der Haar et al., 1993) due to the range of distinct parent materials (glacial, peri-glacial, fluvial, aeolian, organic) and

topography, and thus soil types within a 10-km radius of the university.

The main difference between making peels and monoliths is the location where the soil is impregnated: a peel is impregnated *in situ* and extracted after drying, while a monolith is an undisturbed soil block that is extracted, transported, and then (repeatedly) impregnated in a laboratory (Van Baren and Sombroek, 1981). Monoliths can be created in any soil type, from sands to peats and heavy clays, but is rather time consuming and requires specialized expertise both in the field and in the

laboratory. Their creation and recent methodological developments are rather well described in scientific journals (e.g. Bouma, 1969;Haddad et al., 2009;Allaire and Bochove, 2006;Wessel et al., 2017;Wright, 1971;Donaldson and Beck, 1973;Barahona and Iriarte, 1999;Fitzpatrick et al., 2015), presentations (Fosberg, n.d.) and reports (e.g. Van Baren and Bomer, 1979;Kiniry and Neitsch, n.d.;Day, 1968;Schuurman, 1955), as well as illustrated in online videos and tutorials (e.g. University of Nebraska - Lincoln, 2016;Mueller, 2018). In contrast to soil monoliths, soil peels cannot be made from clay or peat soils since these are

often too wet for impregnation in the field. Peels are therefore limited to relatively coarse sediments that retain less water (lower water holding capacity) and allow more rapid impregnation of fixation agents (because of their higher hydraulic conductivity) which is required in field situations. They thereby provide a rapid and accessible alternative to soil monoliths. The lack-film method for creating peels was first developed in the 1930's (Hähnel, 1962;Voigt, 1936;Jahn, 2006). Yet while the use of soil lacquer peels for scientific purposes has been recognized, e.g. to study sedimentological structures (Bijkerk et

al., 2014;Van den Berg et al., 2007), for palaeo-geochemical analysis (Arnoldussen and van Os, 2015) or archaeological applications (Voigt and Gittins, 1977), the guidance available in the scientific literature is scattered, (out)dated and/or incomplete. An English book that stands out is the comprehensive work by Bouma (1969) that details the history of soil and sediment peels as well as a range of fixation agents used to make these peels. Other published work includes a range of Dutch and German-language papers, popular-scientific articles and reports (Vos et al., 2016;Huisman, 1980;TNO, 2010;Van Veen,

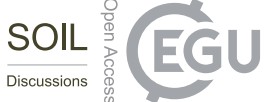

1985;Hähnel, 1961;Voigt, 1936), as well as a few older English-language articles (Voigt and Gittins, 1977;Van Baren and Bomer, 1979;Brown, 1963;Hähnel, 1962). These publications describe a range of materials used to make peels, most notably (nitrocellulose) lacquers but also glue and resin.

The main steps of the published methods for making peels are straightforward: a soil face was prepared under an angle and then (repeatedly) impregnated with a fixation agent, with the peel extracted after drying and then mounted on a wooden board. The challenge of the published methods lies in the fixation agents used 50 years ago that required use of toxic solvents (e.g. acetone, toluene, xylene, thinner; Bouma, 1969) in the field to achieve the right viscosity, increasing the risk of harming people and the environment. The resulting peel was rather fragile, hence reinforcement with cheesecloth or bandage was required to prevent rupture of the dried lacquer peel (Bouma, 1969). This fragility results in a lower preservation potential, which we have noted was especially challenging when the soil peels were frequently used for teaching.

Here we describe and illustrate a more simple, safe, and durable and thereby more accessible approach of making soil lacquer peels, which relies on the use of glue available at hardware stores. While still synthetic, this glue is less harmful than the fixation agents, and its use straight out of the can reduces spilling risk associated with the mixing of chemicals in the field. Finally, this method invariably yields excellent results also when used by those who have received no training. This new twist to an old method was developed by Gert Peek, a soil science educator at Wageningen University who started teaching at what was then the Laboratory for Soil Science and Geology at the Landbouwhogeschool Wageningen. As such, this method was used to collect both teaching material and data for MSc theses (e.g. van der Beek and Ellenkamp, 2003), and to enthuse hundreds of students to get a podzol above their bed, through the organization of 'soil profile weekends'. As we believe it is essential that scientific methods are preserved for future generations, we now report the simple steps to capture the beauty of sandy soils for use in universities, schools, government buildings, museums, or simply at home.

## 2 Taking the soil peel: six main steps

### 2.1 Collect the required materials

*Materials needed.* A range of materials is required to make a soil peel that can typically be found in any hardware store. Table 2 lists all materials required to prepare the soil face (a spade, pruning scissors or garden shears, nail clippers, soil knife, ruler), to secure the soil (glue), to extract the peel (wooden board, spade, soil knife, pruning scissors or garden shears, garbage bag), finish the lacquer peel (glue, notched trowel, Stanley knife, nail clippers, scissors), and mount it (hooks). In terms of personal gear, garden gloves and clothes that can get dirty are sufficient. Any size can be chosen for the final size of the soil peel, and thus the size of the wooden board. Soil profiles at Wageningen University are typically 30×120 cm. A wooden panel > 12 mm thick (to prevent warping) is used for mounting - we use multiplex or MDF though any wood can be chosen, depending on desired aesthetics.

*Characteristics of the glue.* The fixation agent used to impregnate the soil face is a liquid contact adhesive based on neoprene rubber. Originally designed for shoe repairs that requires two sides to be pressed together, this neoprene rubber contact




adhesive works very well for making peels because it is flexible yet strong when dry. This flexibility is key for successful extraction of the peel from the soil face: glue that fully hardens when dry (like wood glue or glues used to impregnate monoliths in the lab) will break upon extraction and/or mounting of the peel. Another benefit of this glue is that it does not shrink when drying, unlike the lacquer used for instance by (Hähnel, 1962). In the Netherlands, neoprene rubber contact adhesive is sold

as BisonKit Universal (Bolton Adhesives, Rotterdam, The Netherlands (Bison International, 2018b)), which is internationally sold by the same manufacturer under the brand names Uhu Kontakt Kleber and Griffon Contact. The yellowish brown color this glue does not affect the colour of the final peel. Neoprene rubber contact adhesive is also known as polychloroprene glue, contact cement, or contact adhesive, and is elsewhere sold by manufacturers such as 3M, DAP Weldwood, Pliobond and K-Flex-USA - check the suitability of these products in the field before purchasing large volumes. Because some of these brands

still contain toluene, it is also advisable to request (material) safety data sheets (known as (M)SDS in the USA) to check for any required personal protective equipment.

The volume of glue ($V_G$ [L]) required to make a peel, including excess edges and mounting the peel, is calculated as Equations 1 and 2:

$$V_G = 3.8 \times (b_W + 0.2) \times (b_L + 0.2) + V_M \qquad \text{Eq. 1}$$

$$V_M = 0.1 \times b_W b_L \qquad \text{Eq. 2}$$

where $b_W$ [m] and $b_L$ [m] are the width and length of the wooden board and thus the final size of the peel, respectively, and $V_M$
[L] is the volume of glue needed to mount the peel to the wooden board. For a final peel size of $30 \times 120$ cm, 3 L is sufficient. At a cost of 5-20€ per L, the total costs of a typical profile amount to under 75€ As many stores allow return of unopened cans of glue, we typically purchase more glue than we need and return the excess.

## 2.2 General preparation

*Find a good location.* In the old days (up to the 1990's) when workload at universities was still low, the frequent and lengthy
soil mapping field courses allowed for many opportunities to find beautiful soils and capture them in peels. Finding a good location can just be a factor of being outside a lot, knowing the surroundings, and scraping off the outer few centimeters of an exposed road cut to reveal the original soil underneath. Alternatively, with less time spent outside, good locations can also be found using digital maps that are often available online. Whether outside or behind a computer, four main factors determine the suitability of a location for making a soil peel: 1) soil texture, 2) groundwater depth, 3) a natural or man-made elevation
difference, and 4) accessibility (Fig. 2a).

First, regarding *soil texture*, lacquer peels are best made in unconsolidated sandy deposits (such as commonly found in delta areas) with low clay, silt and organic content and ideally low rock fragment or gravel content. Clay and silt have low permeability (Rawls et al., 1982) and so does organic matter when compacted (Ohu et al., 1985), and thereby result in very



shallow impregnation of the glue, causing potentially fragile layers. Based on years of field experience making soil peels we found that the textural classes "sand" and "loamy sand" (Soil Science Division Staff, 2017) are best suitable. This indicates that the clay + silt content should not exceed 30%, with a maximum of 15% clay. The minimum sand content should therefore be 70%. At the same time, the organic matter content should not exceed ~8% (humic conditions, sensu De Bakker and

Schelling, 1966). Rock fragments and gravel are challenging to work with because they affect the smooth preparation of the soil face (Section 2.3) and additionally may fall off the final peel (Section 2.6), although results can still be quite successful. To find locations with suitable soil texture and organic matter content, the S-World model (Stoorvogel et al., 2017) and the SoilGrids tool (ISRIC, 2018b;Batjes, 2012) are both valuable and free resources.

Second, *groundwater depth* is important because results are best if soils are dry, since the glue used does not adhere properly

when soils are wet. Groundwater level variation can be part of hydrological monitoring setups, but also be recorded on soil maps as average highest and lowest groundwater levels (e.g. BIS Nederland, 2018). Given that warm and dry weather in late Spring or Summer are often most beneficial for making soil peels, the most relevant groundwater information there is the average lowest groundwater level (which occurs in Summer). The global map of groundwater table depths created by Fan et al. (2013), albeit coarse, can give a first indication of whether a region may be suitable for making soil peels. Subsequent

combination of soil texture, organic matter, and groundwater information can then provide insight into where peels can be made (e.g. Fig. 3). Combined with information about capillary rise (~2.5 cm in gravel to > 1 m in silt; Singhal and Gupta, 1999), locations of suitable dryness can be found, which is in soils and sediments above the capillary fringe.

Third, an *elevation* difference is essential when making lacquer peels of vertical cross sections of soil or sediment. This elevation difference can be created by digging a soil pit, which can be done by hand. Approximately 1 m$^2$ is needed to have

sufficient work space, with a 1.0-1.5 m depth of the pit to obtain a 0.8-1.3 m long peel. However, as digging a pit can be time consuming, the most ideal places to make peels are natural drops in elevation such as eroded river banks, or man-made cases such as road cuts, quarries, construction works (river restoration, cable installation), or archaeological digs. Contact local authorities or companies to ask for temporary opportunities, or consult elevation maps for more permanent locations. Elevation maps are often available online. Digital elevation models (DEM) may also be used, for example the AHN (*Actueel*

*Hoogtebestand Nederland*) in The Netherlands is a freely available elevation map with a resolution of 0.5 by 0.5 m (Van Heerd and Van't Zand, 1999). International examples include the EU-DEM with a resolution of 25 by 25 m (EEA, 2018).

*Arrange permission.* Locate the landowner and ask their permission. As many non-soil scientists do not know what a lacquer peel is, a simple explanation free of scientific jargon is to refer to it as a 'soil painting' or 'soil art'. Be honest about the use of glue, but also explain that you will clean everything up. Check whether the landowner would like to receive notice about the

exact moment the fieldwork is planned – though as the process of making a soil peel is weather-dependent, this can often not be indicated much in advance.

*Get the timing right.* In some climates, planning ahead for making lacquer peels can be challenging as this activity is rather weather dependent. Results are best when soils are dry, creating more intense colours and higher contrast of colours in the peel. In The Netherlands, our experience with the 'soil profile weekends' learned that two weeks of dry weather in late Spring





or Summer is sufficient to achieve good results. We have never had issues with soils that were too dry, and with the materials we use there is no need to spray the soil with water as suggested by Bouma (1969). While it is possible to make a peel when the soil is moist, the result is not as beautiful because of reduced appearance of for instance podzol fibers, or simply because the glue will not adhere to the sand. Note that while soil moisture contents may strongly vary in time, there may also be

considerable differences within a soil profile. When sand may be already dry, horizons with more organic matter or clay can still be quite moist because of their strong effect on soil water retention (Rawls et al., 2003;Wösten et al., 1999). These within-profile differences may be exacerbated by impermeable layers: we once encountered major issues when extracting a peel from a podzol that had a perched water table due to an impermeable Bh horizon. While application of the glue (Section 2.4) was successful, the extracted peel showed that the glue had not adhered to the very wet E horizon above the Bh, while the C horizon

below the impermeable layer was dry and adhered just fine. This peel was later restored in the lab (Section 2.6) using dried sand collected from the E horizon.

Dry weather is recommended both in the couple of weeks before making a peel as well as during the two days in the field (Section 2.3-2.6), when also air temperature is important. Follow manufacturer's recommendations regarding the temperature at which the glue can be used (e.g. 15-25°C, Bison International, 2018a). Particularly the first hours after impregnation are

critical because any rain occurring soon after impregnation (within 6-10 h) may create bubbles in the glue, resulting in poor impregnation and therefore 'bold' spots with reduced sand cover upon extraction. High relative humidity can potentially have similar effects (e.g. >65%, Bison International, 2018a), although we have never had such issues in the field.

## 2.3 Field preparation: prepare soil face and cut all roots

*Prepare soil face.* Use a spade to make a straight soil face at a 65° (loamy sand) to 80° (sand) angle (Fig. 2b). The dimensions

of the soil face to be impregnated should be somewhat larger than the intended size of the lacquer peel. Make the soil face 10 cm wider than the final peel on either side (Fig. 2b), because it is never fully predictable how the glue will flow and thus what the final surface is that will be covered. Also, extend it 15-20 cm below the bottom end of the intended peel to allow unimpeded flow. An additional benefit of making the soil face larger than the final size of the peel is that it allows selection of the best or most beautiful part of the profile for mounting. After all, the final appearance of the front of the lacquer peel remains hidden

until after excavation, as the lacquer peel is a mirror image of the soil face. It is therefore always a surprise what the final peel will look like, which is why having additional space to choose the most beautiful part for installation on the board is useful. The prepared soil face should be as smooth and straight as possible – any bumps and hollows can hamper smooth distribution of the glue in the next step. Perfection is not possible though, especially when sediments are brittle or gravelly. It would not be the first time that removing 'one last thing' can cause collapse of part of the soil face and thereby necessitate much larger

restoration work before the glue can be applied.

*Trim roots and remove rock fragments.* Cut away all roots protruding from the soil face using garden or nail clippers (for large and small roots, respectively) and remove any rocks or large rock fragments (Fig. 2c). Roots or rocks that stick out will retain glue and can thereby create glue-less pockets that will appear as holes in the finished lacquer peel. Cut the roots as close to the





soil face as possible while avoiding any dislocation of sand grains. This can be a rather tedious process as the number of roots can be surprisingly high. Yet careful removal of roots and rock fragments will allow smoother impregnation of the soil face (Step 3), easier mounting of the peel on the wooden board (Step 5), and thus better final results.

*Make ledge.* Create a 5-cm ledge above the soil face (Fig. 2c), providing a place to pour the glue, and preventing any soil

material from above from falling on the profile. If the top of the soil face is the same as the mineral soil surface this ledge can be created by removing any litter and vegetation. If the top of the soil face starts mid-way a slope, this ledge can be made by simply cutting 5 cm into the soil.

*Install gutter.* Position a gutter underneath the soil face that can collect any excess glue (Fig. 2c). A piece of PVC pipe (diameter 10-15 cm) sliced in two and then capped on both ends can function as a good gutter, although a plastic bag may also do if

positioned well. If the firmness of the soil profile allows, cut a 5-cm overhang below its bottom and locate the gutter underneath this overhang.

The total time required for field preparation strongly varies with the degree of care taken when preparing the soil face – a general time estimate for this step is ~2 h for soils with a good rooting pattern. In sediments with no roots this step can be done in 5-10 min.

**2.4 Impregnation: securing the sand grains with glue**

*Apply the glue.* To allow for rapid application of the glue, open all the cans of glue and place them within reach of the soil face – or close to a helping hand who can give the cans to the person applying the glue. In contrast to the previously discussed older methods that required on-site mixing of glues or lacquers with thinning chemicals, glues used here are ready-for-use and can thus be used straight out of the can. Application of the glue on the soil face is easiest when using wide-mouth cans (~15 cm

diameter); we recommend using an empty vegetable can for pouring if the glue container has a more narrow opening. Gently pour the glue by starting at the top ledge and moving the can across the width of the profile in a zig-zag pattern (Fig. 2d). While the glue moves downwards also move down the location where the glue is applied. Most likely, a finger-like pattern will appear in the glue (Fig. 2d, Video 1), especially when the soil face is rather vertical. This preferential flow is caused by the fact that liquids have a strong tendency to follow existing zones of (even slightly) higher liquid content (Liu et al., 1994),

because of the large differences in hydraulic conductivity and thus flow velocity in dry and wet materials. Fill in the gaps between the preferential flow paths by pouring glue at their top and continue this process until the glue has reached the bottom of the soil face. When the bottom of the profile is reached, the impregnation stage is finished. We recommend application of only a single layer of glue: we qualitatively tested the effect of adding additional layers of glue, which did not improve the final product. In one case, application of a second layer of glue even resulted in movement of the initial layer, creating a glue-

less patch and thus a hole in the final peel.

*Clean up.* With the glue application done, the impregnation step of making the lacquer peel is finished. The neoprene rubber contact adhesive is so strong and yet flexible within the first days of application that it can easily hold the weight of a soil profile without tearing. As such, reinforcement of the peel with cheesecloth as directed by Bouma (1969) is not required.





Collect any excess glue that is still liquid from the top ledge and the bottom gutter. Remove all trash and leave the site such that any visitors (people or animals) cannot harm themselves. Cover the impregnated soil face with a large (fisherman's) umbrella if there is a chance of light rain, and wait 20-24 h to let the glue dry (Fig. 2e). The exact drying time will depend on meteorological conditions (air temperature, relative humidity, and wind) and exposure of the profile. It may be that the profile

is dry and ready for extraction after less than 20-24 h. Testing of potentially reduced drying times in different conditions is advisable in cases where time is tight and weather conditions are advantageous.

## 2.5 Extracting the peel

*Clear sides.* Extraction of the peel from the soil face involves the repositioning of a lot of loose sand from behind the peel to

the sides. To facilitate this sand removal, make sure that the soil face on either side of the impregnated section is flush with the lacquer peel for a width of ~15 cm on either side. It is also advisable to remove sand around the bottom of the profile.

*Cut out peel from above.* Everything is now in place to start digging out the peel from above using the serrated edge of a (soil) knife. Starting at the top ledge, use the (soil) knife like a saw to make a cut 5-10 cm away from the glued soil face across the entire length of the peel (Fig. 2f, Video 1). The knife cuts fine roots; use garden clippers to cut off larger roots ~5-10 cm away

from the glue. The further out from the glue you make these cuts, the longer the roots will be that stick out of the finished peel, which can always be trimmed in Step 5.

Extraction of the soil peel is best done with two people, and can be done from the top (as outlined here) or from below (as outlined by Bouma (1969)). To extract the peel from the top, one person cuts away the soil and moves loose sand away from behind the profile and works their way down the profile. Once the top of the lacquer peel has been freed, a second person then

presses a wooden board against the soil face and holds the top of the profile against the board (Fig. 2f, Video 1). This is to support the peel and prevent any tearing along fragile layers such as podzol fibers or thin loam bands. If the peel is heavy, for instance in the case of very structured soil, it can be partly folded over the top of the wooden board. Covering the edge of the wooden board with a thick towel can then reduce the risk of tearing that can occur in fragile layers.

Digging out the lacquer peel can be easy and straightforward if the peel is small and does not contain roots or concretions.

Very small profiles (e.g. 40 by 40 cm) can even be done by a single person. Extracting a more typically sized peel (e.g. 30 cm wide by 120 cm long) is not necessarily difficult but it can be arduous if layers are densely rooted or structured. Still, 15 to 30 minutes is usually sufficient to remove peels from a soil face.

## 2.6 Mounting the peel

*Choose your mounting location.* The extracted soil peel can be mounted on a wooden board either directly in the field, or after

transporting the peel to a laboratory, shed, carport or garage. Mounting the peel in the field allows for safer transport, yet it does typically mean that the size of the wooden board and thus the final size of the lacquer peel is predetermined – unless there is a possibility to bring power tools to the field to trim a board to size. Using a fixed board size is not a problem when making



soil peels for teaching or outreach collection, but when using peels for soil art it can be worthwhile to determine the final peel size after extraction. After all, since the peel is a mirror image of the soil face (as discussed in Step 2), its final appearance remains a surprise until it is extracted from its location.

*Test positioning.* When ready to mount the peel, test its position on the wooden board to decide which features to keep.

Measuring how much the peel will extend beyond the sides of the wooden board helps exact positioning once the board is glued. If the peel is too heavy to lift, reduce its weight by remove large aggregates by hand, and/or by very carefully removing any large clumps soil with a soft brush. A brush may also be used to remove loose sand (always stroke sand away in the direction of any soil layering) but only if the glue is fully dry.

*Glue the wooden board.* Cover the wooden board with some of the remaining glue (Fig. 2g), making sure to particularly cover

its sides and corners as these are the most vulnerable parts of the finished peel. Use of a notched trowel facilitates an even spread of the glue, while corners and sides can be reached by hand using household gloves. Work swiftly as the glue dries quickly, particularly when weather is warm (> 25°C) and windy.

*Attach peel to board.* Lift the lacquer peel up with two people and place it on the wooden board directly in the desired location: as the glue will create an instant grip, changing the alignment of the lacquer peel will be very challenging if not impossible.

*Press peel in place.* Carefully but firmly press the lacquer peel to the wooden board with your fingers. Again pay particular attention to the sides and corners of the wooden board to secure these well.

*Remove loose sand.* Turn the peel on its side and release any loose sand still resting on the peel by manually knocking the back of the wooden panel. Repeat until no sand falls off anymore. Keep some excess material from each layer (soil, any rock fragments, large roots) to restore any damaged patches later if needed.

*Trim peel to size.* Now that the peel has been secured to the wooden panel, trim it using a sharp (Stanley) knife (Fig. 2g). With one side of the knife touching the side of the wood, cut off all parts of the soil peel that extend beyond the wooden board.

*Restore any damaged patches and trim roots (if desired).* In some cases peels may have small holes or damaged patches if glue distribution was not uniform or where rock fragments or larger roots have fallen off. These patches can be easily restored by applying some glue and covering it with the appropriate material for that layer, such as soil particles, a rock fragment, or a

25 large root. This is also the moment where roots can be trimmed if desired using shears or nail clippers. There is no predetermined root length, the final root length is very much part of the artistic freedom and the message that is communicated with the soil peel, if any.

## 2.7 Finishing, installation and maintenance

*Ventilate.* The soil peel now requires some rest in a well ventilated place to let the glue fully solidify – we ventilate our profiles for a minimum of 4 days. As glue fumes can be rather intense, a garage, shed, or covered dry outdoor location is best for this. Make sure to place the lacquer peel in a horizontal position – placing it vertically shortly after mounting may result in vertical movement of the drying glue, and thus distortion of the soil profile.



*Finish.* Many authors suggest impregnating the undisturbed front of lacquer peels (e.g. Huisman, 1980;TNO, 2010) to intensify the colours of the soil particles and secure any loose particles. Our team has done that from 1978 to 2010 using a large can of the cheapest hairspray sold at the local drugstore, applying it one week after the soil peels were mounted. The hairspray did bring out the colours more, but once surprisingly produced such dark colours that any colour variation in the peel was obscured.

It may be that the formula of the hairspray had changed, but the exact reason for this dramatic colour change was unknown. Since then, we do not spray peels anymore, and are very satisfied with the original colours. As such, there was no need to find an alternative impregnation material. In the case that colours are bleak, spraying with hairspray can be a way to intensify colours, but we strongly recommend testing of results along the entire length of the lacquer peel using the trimmed-off edges of the peel. In that case, turn the profile on its side to knock off any loose particles before spraying and ventilate again for a

few days before installation.

*Install.* After a week of rest when the glue will be firm and odourless, the finished lacquer peel can be installed at its final location. Hooks screwed into the top of the board allow hanging it vertically on a wall in a classroom, office, living room, museum, or wherever this piece of science art is desired. If desired, slats can be used to construct a wooden frame around the finished lacquer peel.

*Maintenance*. We have heard reports of people annually impregnating their soil peel with spray to 'maintain its colours'. We have never seen a need for this and do not perform any maintenance of the finished peels. After changing from lacquer to glue, preservation of our peels has improved such that even intensive use in hands-on teaching does not degrade the peels anymore. If required, dust can be carefully removed from between any roots using a vacuum cleaner set at its lowest speed.

## 3 Discussion and conclusion

High participation in the maker-ed and DIY movements (Holtzman et al., 2007;Atkinson, 2006) indicates renewed interest in making things at home, while the potential of visualization is being recognized in science communication and education (Evagorou et al., 2015;Venhuizen et al., in review). At the same time, there is increased interest in the value of soils for life (Keesstra et al., 2016;FAO, 2015). The creation of soil and sediment peels combines all these aspects, and is much easier than many people would think. Materials including glues are readily available at hardware stores, and even novices can create

beautiful peels. Here we discussed the benefits of using peels and the challenges posed by the old methods (e.g. Voigt and Gittins, 1977;Van Baren and Bomer, 1979;Bouma, 1969) used to create these peels. We described the main steps of making a soil peel: impregnation of a smooth soil face with glue in the field before extracting the peel and then mounting it on a wooden panel. Because of a technological advance in the impregnation material (going from lacquers to glue), the method reported here is more safe, simple, successful, durable and accessible because 1) the glue can be used without the use and mixing of

toxic chemicals in the field, 2) the firmness of the resulting peel is such that additional support materials (such as cheesecloth) are not required, and 3) consequently the soil peel will last for a long time, even when intensively used in hands-on teaching. While this method can be applied to a range of moisture contents and sand textures, further research on the best environmental





conditions is required for those interested in achieving perfection in terms of appearance. Similar exploration is advised for alternative glues. Such additional research would be valuable for some (e.g. soil museums), but based on our experience, we believe that those simply interested in capturing a beautiful snapshot of soils can do so with the more qualitative guidance described in this paper. We hope that this thoroughly tested successful and simple method will inspire and enthuse researchers,

educators and the general public to make soil lacquer peels and thereby bring the value and beauty of soils to a wider audience.

**Acknowledgements**

We thank Niels Kijm for assistance with making the lacquer peel for the video instructions, Bob Czaja and Ann Youberg for discussion of the types of glue that are internationally available, and Bison International for information on the characteristics of BisonKit. We furthermore would like to thank the following people for their help in sourcing the data in Table 1: Albert

Bos, Alejandro Becerra, Alessandro Samuel Rosa, Bernd Andeweg, Christine Morgan, Coen Ritsema, Colby Moorberg, Darya van Tienhoven, Erin Bush, Franciska de Vries, Hayley Craig, Ichsani Wheeler, Jacqueline Aitkenhead-Peterson, Jacqueline Hannam, Jakob Wallinga, Jantiene Baartman, Jerry Maroulis, Jetse Stoorvogel, Karen Vancampenhout, Keiko Mori, Kirsten van der Ploeg, Liam Heffernan, Marcos Angelini, Meredith Steele, Michael Strickland, Mirzokhid Mirshadiev, NSSC-GRU, Nynke Schulp, Rachel Creamer, Richard Bardgett, Richard Kraaijenvanger, Stephan Mantel, Stephan Mantel, Wieske

Wentink, Wouter Thijs, Wouter van Gorp and Zhanguo Bai. The authors declare no conflicts of interests. Any use of trade, firm, or product names is for descriptive purposes only and does not imply endorsement by Wageningen University and Research. C.R.S. has received funding from the European Union's Horizon 2020 research and innovation programme under the Marie Skłodowska-Curie grant agreement No. 706428.

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

**Figure 1: Lacquer peels showing (a) paleo podzol (from below inset) covered by drift sands in which a younger podzol is formed, (b) plaggic anthrosol, (c) sedimentary layering, (d) frost wedge, (e) faulting, and (f) colourful sediments. Insets show close-ups of damaged parts of the peel; white horizontal bars represent 10 cm width; panels (a) through (e) are soil and sediment peels made in the Netherlands (Wageningen University collection, The Netherlands), panel (f) is a sediment peel of the Owl Rock member of the Chinle Formation, Chuska Mountains, New Mexico USA (Diné College collection, Tsaile, Arizona).**



### (a) IDENTIFY SUITABLE SITE

▶ Soil texture: sands & loamy sands
▶ Above groundwater table
▶ Exposed soil face/dig a pit
▶ Accessible & get permission
▶ Dry weather (2 weeks)

### (b) PLOT OUT PROFILE

▶ Determine dimensions final peel
▶ Add ±10 cm horizontal margin
▶ Add 15-20 cm bottom margin
▶ Straighten soil face at 65°-80°

*±10 cm margin*

*±15-20 cm margin*

### (c) CLEAN & PREP PROFILE

▶ Take out rocks & trim off roots
▶ Create ±5 cm wide ledge at top
▶ Install gutter

*±5 cm ledge*

*Gutter: PVC pipe, Ø 10-15 cm*

### (d) APPLY GLUE

▶ Start at top ledge
▶ Apply glue evenly over surface
▶ Move horizontally and downward
▶ Excess glue caught by gutter

*Apply glue in S-pattern*

*Excess glue caught by gutter*

### (e) LET DRY

▶ Keep sheltered from rain
▶ Let dry for ±24 hours

### (f) EXTRACT PROFILE

▶ Clear sand away from sides & bottom
▶ From top, cut along margins
▶ Remove soil behind peel in steps
▶ Lift peel from soil face
▶ To prevent damage: press peel against board

*Cut margins*

*Remove peel from top to bottom*

### (g) CUT TO SIZE

▶ Glue extracted peel to wooden board.
▶ Trim to size.

*Trim peel to size*

**Figure 2: Main steps of making a soil peel**



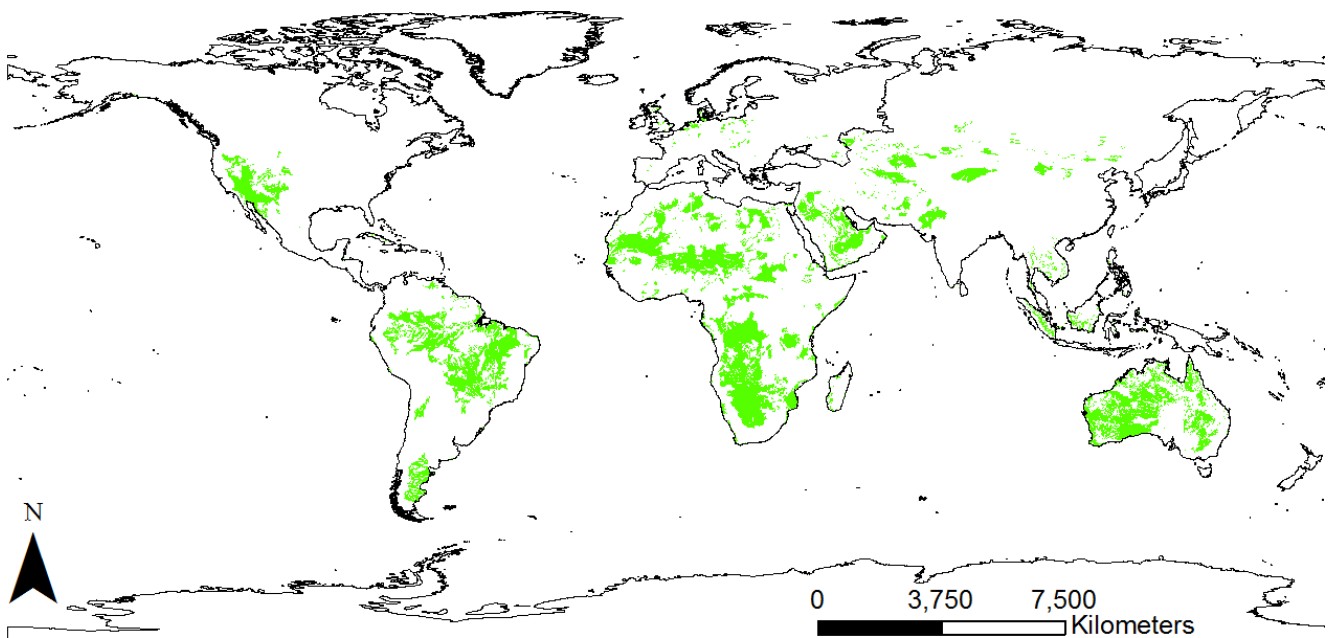

**Figure 3: Overview of locations suitable for making soil peels in the world (sand content > 70%, clay content < 15%, organic matter < 8%, and temporal mean groundwater depth < 1.5 m). This global map was made using soil information obtained from ISRIC (Batjes, 2012;ISRIC, 2018b) and groundwater depth information provided by Fan et al. (2013).**



**Table 1: Example of museums, universities, schools and institutes with soil profile collections (lacquer peels and/or monoliths)**

| Country | Institute | Source |
|---|---|---|
| Argentina | Universidad Nacional de Río Cuarto, Instituto Nacional de Tecnología Agropecuaria | Personal communication Marcos Angelini, Alejandro Becerra. |
| Australia | The University of Sydney (Australian Technology Park) | Personal communication Ichsani Wheeler |
| Austria | University of Vienna | (Feigl, 2016) |
| Belgium | KU Leuven | (ETWIE, 2018); personal communication Karen Vancampenhout. |
| Brazil | Universidade Federal de Lavras, Universidade Federal de Roraima, Universidade Federal de Santa Maria. | (UFRR, 2018;UFLA, 2016;UFSM, 2018). |
| Canada | University of Alberta, University of British Colombia, Great Lakes Forestry Center (Ontario). | (Krzic et al., 2013;Natural Resources Canada, 2018); personal communication Liam Heffernan. |
| Colombia | Museo de Suelos Ciro Molina Garcés, Museo de Suelos Instituto Geográfico Agustín Codazzi. | (UNAL, 2018;IGAC, 2018) |
| China | China Soil Musem; The Modern Soil Monolith Exhibition Center | (GIWSR, 2018;ISSCAS, 2018) |
| Estonia | Soil Museum Estonian University of Life Sciences | (Eesti Maaülikol, 2018) |
| Germany | Halle University | (Jahn, 2006) |
| India | Kerala Forest Research Institute | (Kerala Forest Research Institute, 2018) |
| Indonesia | Museum Tanah (Bogor Soil Museum) | (AMI, 2018) |
| Japan | Natural Museum of History and Science, Natural Resource Inventory Museum, Tsuchino-Yakata, Hokkaido | Personal communication Keiko Mori |
| Netherlands | World Soil Museum, Wageningen University, VU University, Rijksuniversiteit Groningen, HAS Hogeschool, VHL University of Applied Sciences, Museonder, Geologisch Streekmuseum 'de IJsselvallei' | (ISRIC, 2018a;De Hoge Veluwe, 2018;Geologisch Streekmuseum 'de IJsselvallei', 2018); personal communication Bernd Andeweg, Richard Kraaijvanger, Wouter Thijs, Kirsten van der Ploeg. |
| Peru | Museo de Suelos | (Fundacion ILAM, 2018) |





| Russia | Vasily Dokuchaev Museum of Soil Science, St Petersburg; Williams Museum of Soil and Agriculture, Moscow | (Russian Museums, 2018); personal communication Jetse Stoorvogel |
|---|---|---|
| Spain | Universidad de Murcia, Universidad de Granada, Institut Cartogràfic i Geològic de Catalunya | (UM, 2018;UGR, 2018;Lladós et al., 2017) |
| Taiwan | National Taiwan University, Taiwan National Research Institute | (Chen, n.d.;Churchman and Landa, 2014) |
| Thailand | Soil Museum Bangkok | (Thai Museums Database, 2018) |
| United Arab Emirates | Emirates Soil Museum | (Emirates Soil Museum, 2018) |
| United Kingdom | Cranfield University | Personal communication Jacqueline Hannam |
| United States of America | Kansas State University, University of Idaho, Texas A&M, Virginia Tech, West Virginia University, University of Georgia, Smithsonian's National Museum of Natural History (2008-2009), Cayuga Nature Center (NY), Diné College (AZ), American Museum of Natural History (NY) | (Univeristy of Idaho, 2018;Megonigal et al., 2010b;PRI, 2018;American Museum of Natural History, 2018;Fitzpatrick et al., 2015); personal communication Colby Moorberg, Christine Morgan, Meredith Steele. |
| Uzbekistan | State Research Institute of Soil Science and Agrochemistry | (YGK, 2018); personal communication Mirzokhid Mirshadiev |



**Table 2: Materials required and their purpose**

| Material | Purpose | 1. General preparation | 2. Field preparation | 3. Glueing | 4. Peel extraction | 5. Mounting | 6. Finishing |
|---|---|---|---|---|---|---|---|
| Ruler, measuring tape | To stake out the lacquer peel dimensions | | X | | | | |
| | To determine the finished peel dimensions | | | | | X | |
| Spade, shovel | To make a smooth soil face | | X | | | | |
| | To clean up excavated sand | | | | X | | |
| Soil knife (Nisaku Horihori weeding knife, Tomita Cutlery Co. Ltd., Koseki Tsubame-si Niigata, Japan (alternative: large serrated knife with a nice big handle) | To shape the ledge | | X | | | | |
| | To dig out the peel after the glue has dried | | | | X | | |
| Garden clipper/pruner | | | | | X | | |
| Nail clippers (2x) | To cut small roots | | X | | | X | |
| Polychloroprene glue | To secure the soil particles | | | X | | | |
| Garbage bag, pvc pipe sliced in half | To construct a collection unit to capture excess glue | | X | X | | | |




| | | | | | | | |
|---|---|---|---|---|---|---|---|
| Sturdy garbage bags or bucket | To transport empty glue containers (potentially sticky), excess glue | | | X | | | |
| | To transport cut off lacquer peel | | | | X | X | |
| Notched trowel | To evenly spread glue on wooden board | | | | | X | |
| Stanley knife | To cut off all parts of the lacquer peel that extend beyond the wooden board | | | | | X | |
| Workers gloves | Protect hands during digging etc. | | X | X | X | | |
| Latex gloves | Protect hands while glueing board | | | | | X | |
| Wooden board | To support extraction and transport of lacquer peel | | | | X | | |
| | To mount lacquer peel on | | | | | X | |
| Blanket, cloth | To prevent lacquer peel from breaking | | | | X | | |
| Hooks | For mounting on wall | | | | | | X |

**Video 1: Instruction video showing how to make a soil peel in the field**

(uploaded)