# Peer review of "Soil lacquer peel DIY: simply capturing beauty"

_SOIL, 2018_

## Referee Comment (RC1) · Anonymous Referee #1 · 18 Dec 2018

This article shows how to do a soil lacquer peel in the most easy way and with advantages with respect to previous methodologies, it is explained in a very detailed manner to can be do it by yourself. It shows also the potential of this kind of work to show soil processes for educational purposes and to disseminate the importance of soil to society. The fact that to make they use strong though flexible contact adhesive, the potential preservation of the peel with time, and the DIY character do this proposed methodology very attractive for soil scientists and soil and earth-sciences educators. Being very interdisciplinary fit very well with the scope of SOIL. There is a good revision also of the State-of-the-Art of this topic in the introduction. The video uploaded is very useful also. My congratulations to the authors for the contribution to the educational soil science community. In summary I recommend the publication as a Short-Communication

in SOIL.

I have only some few comments or suggestions to improve the manuscript.

In the first part of the introduction to complete I suggest to mention that recently, 2015 was declared the International Year of Soils by the FAO and also that we are currently in the International Decade of Soils declared by the International Union of Soil Sciences.

In the page 2, line 30, correct palaeo-geochemical by paleo-geochemical

In the page 3, line 29, what does it mean MDF?

In the page 16, Figure 2 (f), the picture is not clear what is showing, I suggest to replace to a better one

In the page 19, Table, line of Spain, I miss also the Museo de Historia Natural in Santiago de Compostela where there is a permanent room dedicated to the Soil http://www2.usc.es/museohn/visita_virtual/ .

Sincerely

---

## Referee Comment (RC2) · Anonymous Referee #2 · 18 Jan 2019

Stoof et al. Soil lacquer peel DIY: simply capturing beauty.

General comments

This is a neat paper that concisely describes a methodology to preserve and display soil profiles of sandy soils using a simple lacquer peel approach. These peels are an alternative to monoliths and reveal the often unseen soil profile and are useful tools in teaching, outreach or extension. The step-by-step methodology in the paper provides the opportunity for non-specialists to also create these visual copies of soil profiles. The video is a great supplement that clearly illustrates the methodology. You could also highlight the benefit of monoliths or peels to increase the accessibility of studying soil profiles for people or students with disabilities that prevent them from observing soil in situ (e.g. insert some text around line 10). Some additional text on other kinds of

data or interpretations to select a site would be useful to ensure that the methods can truly be undertaken by non-specialists. Some suggestions are detailed in the minor comments.

Minor comments

P1 line 27 replace 'capture soils' with 'capture soil profiles'

P2 line 16 change 'where the soil is impregnated:' to 'where the soil is impregnated with a fixing agent:'

P3 line 10 add 'handled when' after 'frequently'

P3 line 13 add 'previous' before 'fixation'

P3 line14 You haven't evaluated the expert vs non-expert outcomes so alter this sentence to ' The methods can be easily deployed by those who have received no training'

P4 eq. 2 x is not defined – should this be multiply?

P4 line 24 First sentence- whilst this statement may be true is it necessary to 'find a good location'?

P5 line 8 Also include a concluding statement soil texture should be determined for the final identified location – either by consulting more detailed maps or data or by determining soil texture directly in the field. This ensures compliance with the texture criteria and greater success for the peel. This is particularly important if you are encouraging non-specialists to use the methods as they may use the coarse scale data to derive a suitable site location that locally has different properties.

P5 line 10 Data from hydrological monitoring may be available for specific sites but would not necessarily guide whether high groundwater would be present at the site when searching for suitable locations. Perhaps you can include in another sentence other datasets that can indicate high groundwater e.g .different soil types (e.g. Fluvisols) or landforms (e.g. river terraces, floodplains, former wetlands) that would likely

have high groundwater and should be avoided.

P5 line 18 Replace 'elevation difference' with 'exposure'. It is a vertical section of soil rather than an elevation (which is a height above a datum) that is required here.

P5 line 24 I'm not sure that consulting a DEM would be useful for locating sites to find suitable exposures. Satellite imagery may be more useful to identify fluvial landforms and quarries.

P5 line 27 Could you also comment on co-production of the peels with landowners? The landowner may also like to be involved in co-creating the soil peel and that it may also be a useful resource for them too? Showing examples or sending pictures of peels helps to communicate the process of creating a peel. Also make sure that the landowner is included in any acknowledgements of material produced from the peels.

P6 First paragraph. Did you take any soil moisture measurement at the time of the peels? This might provide useful 'rule-of-thumb' moisture contents to produce the best results (e.g. maximum moisture contents above which peel efficacy is reduced)

P6 line16 replace 'bold' with 'bald'

P8 line 13 replace 'away from' with 'behind'

P8 line 16 replace '...Step 5' with 'Step 5 section 2.6'

P8 line 20 replace '..and holds the top of...' with ' ..that supports..'

P9 line 20 Add sentence e.g. Retain strips to test of additional fixing agents (Section 2.7, Step 2)

P10 line 7 replace 'bleak' with 'weak'

P10 line 23 change '...and it much easier than people would think' to '..can be achieved by non-specialists'.

Figure 2 c) use 'prepare' instead of 'prep'

[Figure]

---

## Author Comment (AC1) · 25 Feb 2019

Thank you for your enthusiastic review. We will incorporate your suggestions about the International Year and Decade of Soils and clarify what we mean by MDF (medium density fibre board) and find a better picture (Fig. 2f). Regarding palaeo vs paleo, we follow the more common spelling 'palaeo'. If the editor prefers 'paleo' we are naturally happy to change this. Finally we thank you for the information about the museum in Santiago de Compostela. We will contact them to ask if they have a monolith or lacquer peel collection (this is not immediately clear from the website) and include them in our list if they do.

---

## Author Comment (AC2) · 25 Feb 2019

Thank you for your positive and constructive review. Your suggestion to include the potential for using soil peels for people with disabilities is great, we will add this to the manuscript. We appreciate your suggestions to clarify the site selection to non-experts, we will incorporate these into the revised manuscript and also process the very good text suggestions. Particularly the texture testing and landscape clues regarding soil wetness are good additions. We do not have moisture content measurements for the soil above the impermeable Bh horizon but since the water table was perched we can more explicitly mention that the soil was saturated. Finally, regarding working with land owners: we typically do not co-create lacquer peels with land owners but the two times we did there was a lot of interest which opened up new research possibilities too. We

will include some words on this in the revised manuscript.

---

## Author Response (AR1)

Dear dr Quinton, hello John,
Please find below the revised annotated manuscript in which we processed the changes suggested by the two anonymous reviewers. All changes were incorporated except for the listing of the Museum in Santiago de Compostela that did have a section on soils but we were not able to verify that this section contained monoliths or
5 lacquer peels. All other changes have been processed (see track changes below).
We additionally listed several other institutes with preserved soil profiles that were new to our list in Table 1.
Thank you for the extra time granted to incorporate these changes,
Kind regards,
Cathelijne Stoof on behalf of the coauthors

[revised manuscript text omitted]

**Video 1: Instruction video showing how to make a soil peel in the field**

(uploaded)